# Tamarind Seed Polysaccharide Hydrolysate Ameliorates Dextran Sulfate Sodium-Induced Ulcerative Colitis via Regulating the Gut Microbiota

**DOI:** 10.3390/ph16081133

**Published:** 2023-08-10

**Authors:** Kangjia Jiang, Duo Wang, Le Su, Xinli Liu, Qiulin Yue, Song Zhang, Lin Zhao

**Affiliations:** 1State Key Laboratory of Biobased Material and Green Papermaking, School of Bioengineering, Qilu University of Technology, Shandong Academy of Sciences, Jinan 250353, China; jiangkangjia0813@163.com (K.J.); wangduogongzuo@163.com (D.W.); sule@sdu.edu.cn (L.S.); vip.lxl@163.com (X.L.); yueqiulin88@163.com (Q.Y.); 2Shandong Chenzhang Biotechnology Co., Ltd., Jinan 250353, China

**Keywords:** ulcerative colitis, tamarind seed polysaccharide hydrolysate, anti-inflammatory effect, short-chain fatty acids, gut microbiota

## Abstract

(1) Background: Ulcerative colitis (UC) is a disease caused by noninfectious chronic inflammation characterized by varying degrees of inflammation affecting the colon or its entire mucosal surface. Current therapeutic strategies rely on the suppression of the immune response, which is effective, but can have detrimental effects. Recently, different plant polysaccharides and their degradation products have received increasing attention due to their prominent biological activities. The aim of this research was to evaluate the mitigation of inflammation exhibited by tamarind seed polysaccharide hydrolysate (TSPH) ingestion in colitis mice. (2) Methods: TSPH was obtained from the hydrolysis of tamarind seed polysaccharide (TSP) by trifluoroacetic acid (TFA). The structure and physical properties of TSPH were characterized by ultraviolet spectroscopy (UV), thin-layer chromatography (TLC), fourier transform infrared spectroscopy (FT-IR), and High-Performance Liquid Chromatography and Electrospray Ionization Mass Spectrometry (HPLC–ESI/MS) analysis. Then, the alleviative effects of the action of TSPH on 2.5% dextran sodium sulfate (DSS)-induced colitis mice were investigated. (3) Results: TSPH restored pathological lesions in the colon and inhibited the over-secretion of pro-inflammatory cytokines in UC mice. The relative expression level of mRNA for colonic tight junction proteins was increased. These findings suggested that TSPH could reduce inflammation in the colon. Additionally, the structure of the gut microbiota was also altered, with beneficial bacteria, including *Prevotella* and *Blautia*, significantly enriched by TSPH. Moreover, the richness of *Blautia* was positively correlated with acetic acid. (4) Conclusions: In conclusion, TSPH suppressed colonic inflammation, alleviated imbalances in the intestinal flora and regulated bacterial metabolites. Thus, this also implies that TSPH has the potential to be a functional food against colitis.

## 1. Introduction

Ulcerative colitis (UC) is a disease in which persistent tissue damage occurs as a result of inflammation affecting the colon or its entire mucosal surface to varying degrees [1]. UC was first described in 1859 [2]. The prevalence of UC continues to rise worldwide, particularly within Western countries [3]. For example, in 2018, the prevalence of UC in Canada is estimated to be 0.725%, while by 2030, the prevalence is projected to be 1% [4].

Until now, the pathogenesis of UC has not been thoroughly investigated. However, in clinical and experimental studies, it is often mentioned that there is a non-negligible relationship between the gut microflora and the host body’s inflammatory response [5,6,7]. Modern clinical studies are beginning to utilize new sequencing technologies to analyze microbiota that differs significantly in the gut of UC patients and healthy individuals. An increasing number of studies have found that people with multiple sclerosis typically have changes in the structure of the flora in their intestines [8,9,10]. These imbalances in the intestinal flora can damage the intestinal mucosa, compromising barrier integrity, and more microbiota cross the barrier, activating the immune response and increasing the incidence of various diseases [11]. Therefore, intestinal microbiota disturbances are considered an essential cause of UC [12].

Under normal conditions, the intestinal microbiota supports the biotransformation of many compounds [13]. The intestinal flora metabolizes a variety of complex nutrients, ultimately synthesizing and fermenting them to form short-chain fatty acids (SCFA), including acetic, propionic, and butyric acid. SCFAs can simultaneously replenish energy and nutrients to the host and microorganisms [14]. In addition, SCFA modulates the lung immune environment and reduces inflammation by activating G protein-coupled receptors [15]. The dominant SCFAs in the colon are acetic acid, which is central to the carbohydrate and fat metabolic pathways [16]. Hung et al. found that acetate treatment suppressed overexpression of nuclear NF-κB p65 in lung tissue and improved acute lung inflammation [17]. Nakano et al. demonstrated that acetate ameliorates damage to mouse colon epithelial cells by activating the JNK and Rho signaling pathways [18]. Butyric acid protects the intestinal mucosa from damage and maintains integrity [19]. Some studies have reported that propionic acid improves autoimmunity and reduces neurodegenerative diseases [20]. SCFA also lowers the intestinal tract’s pH, favoring the colonization of bacteria such as *Lactobacillus* and *Bifidobacterium*. These beneficial gut bacteria ferment and produce SCFAs in large quantities, which significantly maintain a healthy immune response [21].

Many factors can influence the production of SCFA, including the available fermentation substrates [22]. Iwaya et al. found that n-butyric acid levels were markedly elevated in the cecum contents of mice ingesting short-isomerized oligosaccharides (S-IMO) with DP = 3.3, effectively delaying the development of UC. In contrast, no remission of UC was observed in mice fed long iso-oligosaccharides (L-IMO) with DP = 8.84. The concentration of butyric acid was also increased after treatment with S-IMO, which favored the growth of probiotics [23]. Therefore, polysaccharides or oligosaccharides are key factors that influence the structure of the intestinal flora.

Recently, various natural gum polysaccharides of plant origin have been investigated and used in different types of drugs [24]. It is because they are characterized by high stability, bioavailability, broad accessibility, non-toxicity and reasonable price [25]. Among the various polysaccharides of plant origin, tamarind seed polysaccharide (TSP) is a promising biopolymer [26]. Its structure consists of glucose, xylose, and galactose in the ratio of 2.8:2.25:1.0 [27]. In their GRAS announcement 2014, the U.S. Food and Drug Administration (FDA) identified this TSP as a substance that is generally recognized as safe [28]. Therefore, it is often used as a cosmetic material, food additive and chemical material [29]. It has been previously reported in the literature that the polysaccharide obtained from tamarind seeds showed immunomodulatory activities, such as enhancing the phagocytic ability of neutrophils [30]. Aravind et al. used the tamarind seed polysaccharide (PST001) to treat different cell lines from humans and mice. They found that PST001 has immunomodulatory and tumor-suppressive activities and could potentially be developed as an anticancer drug [31]. However, TSP is a non-starchy polysaccharide that cannot be digested by animals and will reduce the utilization of nutrients by increasing intestinal viscosity [26,32]. Therefore, in order to eliminate the antinutritional effect of TSP and to take into account its biological activity, we used trifluoroacetic acid (TFA) to hydrolyze TSP and investigated the structure of acid hydrolysis product of TSP (TSPH) and its alleviating effect on colitis.

## 2. Results

### 2.1. Structural Composition Analysis

UV spectra was used to initially analyze the purity of TSP and TSPH. TSP had a small absorption peak at 260–280 nm, which indicated that it contains proteins. Moreover, the absence of an absorption peak of TSPH at 260–280 nm in the UV spectrum indicated that it does not contain protein (Figure 1A).

### 2.2. TLC Analysis

According to the results of TLC, TSPH was mainly composed of saccharides with different degrees of polymerization (DP) (Figure 1B).

### 2.3. FT-IR Spectra Analysis

Figure 1C shows that the FT–IR of TSP and TSPH followed roughly the same trend, but some characteristic peaks in the spectrum of TSPH were significantly different from those of TSP in terms of intensity. There was a stretching peak at 3312 cm^−1^ in the TSPH spectrum, which belonged to the stretching vibration of the O–H bond, probably due to the breakage of the glycosidic bond during acid digestion, which exposed more hydroxyl groups inside TSPH. The peak at 2929 cm^−1^ occurs due to the stretching vibration of C–H. The peak at 1671 cm^−1^ was assigned to the bending vibration of O–H in the sugar ring [33]. The absorption peaks at 1425 cm^−1^ and 1376 cm^−1^ belonged to the deformation vibration of C–H, including the CH_2_ and CH_3_ groups. The peak at 1201 cm^−1^ belonged to the characteristic absorption peak of the C–OH of the side group of the sugar ring. The peaks at 1138 cm^−1^ and 1020 cm^−1^ were caused by the C–O vibrations of the hydroxyl groups on the pyranose ring. And their vibrational peaks were significantly stronger in TSPH compared to TSP, indicating that TSPH contains more pyranose [34]. The peak at 871 cm^−1^ was the characteristic absorption peak of the β-type glycosidic linkages [35].

### 2.4. High Performance Liquid Chromatography and Electrospray Ionization Mass Spectrometry (HPLC–ESI/MS) Analysis

To study in depth the composition and structure of the hydrolysis products obtained from TSP, molecular weight and possible chemical groups were determined by ESI-MS spectroscopy in this study. Based on the results of the seven major peaks in the HPLC chromatogram (Figure 2A), ESI-MS analysis further illustrated their structures (Figure 3B). The peak *m*/*z* 661, 823 in Figure 2B-a were considered to be [M_3–4_ + 2PMP − H_2_O + H]^+^ loss of one 1-phenyl-3-methyl-5-pyrazolone (PMP) group. The peak *m*/*z* 337 in Figure 2B a,b was considered to be [M_2_ + 2PMP − H_2_O + H]^+^ loss of a galactosyl residue. Figure 2B b–g shows that the ions at *m*/*z* 511, 673, and 835 can be regarded as [M_1–3_ + 2PMP − H_2_O + H]^+^ ion peaks, respectively [36]. The peak *m*/*z* 175 was considered to be [PMP + H]^+^ in Figure 2B-d. According to ESI–MS data (Figure 2B-f), [M + 2PMP − H_2_O + H]^+^(*m*/*z* 481) ions indicate the presence of xylose structural units. The peak at *m*/*z* 363 in Figure 2B-g indicates that the dimer is oxidized to an aldehyde or ketone upon addition to sodium [37]. Taken together, the results indicated that TSPH was composed of saccharides with DP 2–4 and monosaccharides.

### 2.5. TSPH Ameliorated Colitis Symptoms in DSS-Treated Mice

During 0–15 days, normal mice showed a slow increase in body weight (Figure 3A,B). However, starting on day 5, the administration of 2.5% DSS (*w*/*v*) resulted in mice experiencing diarrhea/bloody stool excretion, body weight loss, and a gradual increase in DAI values. On day 15 of the experiment, mice in the DSS group showed a 12.78% body weight loss compared to the NOR group. However, the treatment of UC mice with TSPH at different doses induced an increase in body weight and a meaningful reduction in the DAI values.

As the results in Figure 3C,D show, the colon of normal mice was significantly shortened due to the intervention of DSS (*p* < 0.01). Furthermore, this unfavorable change was alleviated by treatment with TSPH-L and TSPH-H (*p* < 0.01).

### 2.6. TSPH Reduced the Histological Injury of Colon

Under H&E observation, the normal colon showed a good mural structure with no loss of crypt cells. However, the colon of DSS-treated mice showed infiltration of inflammatory cells and disruption of the crypt structure, accompanied by irregular villi surfaces. TSPH treatment ameliorated these damages and suppressed inflammation (Figure 3E). This conclusion was also confirmed by histological scoring values (Figure 3F).

### 2.7. TSPH Inhibited Inflammatory Cytokine Secretion

The result was shown in Figure 4A, compared with the normal mice, the concentration of IL-1β, IL-6, and TNF-α in the DSS group were increased by 89.11%, 64.67%, and 71.58%, respectively (*p* < 0.01). The results of the above mouse serum indicate that a severe inflammatory response occurs in the body of the mouse. Treatment of colitis mice with different concentrations of TSPH showed better results. The concentration of IL-1β, IL-6, and TNF-α in the low-dose TSPH group were decreased by 53.52%, 60.44%, and 71.11%, respectively (*p* < 0.01). This apparent remission result was equally observed in the examination of colonic tissue (Figure 4A). The above results showed that TSPH markedly inhibited the further exacerbation of colitis symptoms in mice.

### 2.8. TSPH Regulates Aberrant Expression of Tight Junction(TJ) Proteins

Compared with the NOR group, DSS significantly decreased the transcript-level expression of ZO-1 and Occludin by 52.99% and 85.83% (*p* < 0.01), respectively (Figure 5A,B). In colitis mice, the relative mRNA expression of ZO-1 was notably enhanced in the groups treated with 2 and 4 g/kg of TSPH (*p* < 0.01). Data from Western blot also confirmed this result (Figure 5C,D).

### 2.9. TSPH Promoted the Production of SCFAs

Acetic acid content was reduced by 44.07% in the DSS group compared to normal healthy mice (*p* < 0.05). However, the acetic acid in the TSPH-L and TSPH-H groups was 54.10%, 59.14% higher than those in the DSS group, respectively (*p* < 0.01). Surprisingly, TSPH had no significant effect on butyric acid content in colitis mice (Figure 6).

### 2.10. LBGH Regulated Gut Microbiota

Rarefaction analysis showed that the sequence depth of the gut microbial environment was adequately captured in each sample. The results showed that the data was adequate to reflect the biological information and was suitable for further analysis (Figure 7A). The results presented in Figure 7B,C indicate that there were differences in colony structure between the groups. In addition, the NOR group was farther away from the DSS group, suggesting a greater difference between the two groups, which was partially reversed by TSPH. As shown in the results of the composition of gut microorganisms in phylum and genus, *Bacteroidetes*, *Firmicutes*, *Verrucomicrobia*, *Proteobacteria* and *Actinobacteria* were the five most dominant phyla in the microbiota of mice with colitis (Figure 7D,E).

*Prevotella* and *Ruminococcus* in the DSS group were the most abundant of all groups. *Lactobacillales* and *Desulfovibrio* in the TSPH-L group were the most abundant of all groups, and *Akkermansia* and *Blautia* in the TSPH-H group were the most abundant of all groups (Figure 8A).

To further reveal the specific genera of intestinal flora affected by TSPH, the genera in the cecum contents flora were analyzed. The data in Figure 8B showed that DSS intervention was responsible for a marked reduction in the content of *Prevotella* and *Blautia* in the intestines of a normal mouse. However, the levels of *Akkermansia* and *Coprobacillus* increased markedly. Compared to the DSS group, mice of the TSPH-H group displayed a marked increase in *Prevotella* and *Blautia* but a decrease in *Coprobacillus* (*p* < 0.05). Interestingly, high doses of TSPH markedly increased the amounts of *Akkermansia* even beyond that of the NOR group (*p* < 0.05).

In Figure 9, there was a positive association between SCFAs (including acetic, propionic and butyric acids) and *Blautia*, but negatively correlated with *Roseburia* and *AF12*. IL-1β and TNF-α in colon tissues were significantly positively correlated with *Coprobacillus* but negatively correlated with *Prevotella* and *Blautia*.

## 3. Discussion

In previous studies, TSP was identified as consisting of a β-D-(1, 4)-glucan backbone with branched chains of galactose and xylose, belonging to the group of galactoxyloglucan [38]. Aravind et al. found that galactoxyloglucan possesses significant antitumor activity and modulates immune functions [39]. Here, we reported a new hydrolysis product of TSP by TFA, TSPH, consisting of monosaccharides and oligosaccharides (DP = 2–4). Based on the components and previous related investigations, we investigated the mitigating effect of TSPH on colitis and its regulatory properties on the gut microbiota.

Excess TNF-α has been reported to enhance local or systemic inflammation, disrupt tight junctions, leading to worsening colitis [40]. High levels of typical pro-inflammatory cytokines (IL-6 and IL-1β), are involved in increased inflammation and histopathology [41]. The data showed that TSPH intake markedly suppressed the elevated levels of pro-inflammatory cytokines in UC mice (*p* < 0.05). Increasing evidence suggests that part of the immune response and inflammation occurs as a result of dysregulation of the NF-κB inflammatory pathway, which accelerates the secretion and accumulation of pro-inflammatory cytokines, which involves almost all diseases (including colitis) [42,43]. These studies provide an essential research direction for further exploring whether TSPH exerts anti-inflammatory effects by regulating the NF-κB pathway.

As a result of alterations in tight junctions, the intestinal barrier is disrupted, allowing more microorganisms and their metabolites to approach and pass through the epithelial barrier, accelerating the progression of colitis [44]. Occludin is a major component of the TJ chain backbone and interacts with ZO-1 [45,46,47]. Our results showed that TSPH intake remarkably restored the tight junctions of the colonic tissues and recovered the intestinal barrier function. Data from histologic evaluation of the colon also suggested this conclusion.

The intestinal environment includes microbiota and its metabolites. Once the intestinal environment is imbalanced, it can further lead to pathologies in the intestines [48,49,50]. Numerous surveys have shown that microbiota members derive most of their carbon and energy from dietary carbohydrates of plant and animal origin [51]. Therefore, we suggested that polysaccharides are crucial in regulating the intestinal flora. On the basis of gene sequencing of cecum contents, we analyzed the composition of the mouse gut microbiota. As we hypothesized, TSPH could alter structural components of the intestinal microbiota, restoring the abundance of *Prevotella*, *Blautia*, and *Akkermansia* and decreasing the abundance of *Coprobacillus*.

Furthermore, from the correlation analysis, we observed a negative correlation between *Prevotella* and inflammatory factors in colonic tissue. Previous studies have shown that *Prevotella* alleviates intestinal inflammation and that the number of *Prevotella* is reduced in UC patients compared to healthy patients [52,53]. In addition to the above bacteria, we observed that high doses of TSPH could enrich the abundance of *Akkermansia*. *Akkermansia* has been reported to degrade intestinal mucin and produce SCFAs to alleviate colitis [54].

According to previous studies, *Prevotella*, *Blautia*, and *Akkermansia*, which are increased gut microbes, may induce alterations in metabolites [55]. In our study, acetic acid, one of the main metabolites, was recovered by TSPH even more than the NOR group. It is interesting to note that this result may be endorsed by *Blautia*, which was enriched by TSPH and identified as a producer of acetic acid [56]. Based on the correlation analysis, the results showed a positive association between the levels of *Blautia* and acetic acid. We reviewed several papers and found that acetic acid significantly inhibited LPS-induced phosphorylation levels of specific proteins in the NF-κB pathway in IEC-6 cells [57]. This indicated that acetic acid produced by intestinal microorganisms may be a major inhibitor of colitis.

In summary, we concluded that TSPH has the biological activity to reshape the intestinal microbiota. On the one hand, TSPH could improve the abundance of *Blautia* and *Akkermansia*. Proliferation of beneficial gut microbiota enhanced the amount of protective SCFAs (such as acetic acid), which stimulates anti-inflammatory effects and enhances barrier protection, thereby attenuating intestinal injury [58]. On the other hand, TSPH also reduced the number of pathogenic bacteria such as *Coprobacillus* to suppress colitis.

## 4. Materials and Methods

### 4.1. Hydrolysis and Purification of Oligosaccharide

TSP was purchased from Tianjin Anwen Hydrosol Technology Co., Ltd. (Tianjin, China). A total of 1.0 mg of TSP was degraded in 20 mL of 0.4 mol/L TFA (90 °C, 2 h), then centrifuged (3000× *g*, 10 min) [59]. The supernatant was dried (60 °C, 15 kPa) in a vacuum drying oven (DZF-6020, Jinghong, Shanghai). Added 10 mL of methanol to the solid sample, dissolved and mixed well, then dried again and repeated 3–5 times until the TFA was removed entirely. After dissolving the solid samples with appropriate amount of distilled water, the obtained solution was placed in a freeze dryer (−30 °C, 4 Pa vacuum) for lyophilization (Gizs-1, Kaizheng, Changzhou, China). TSPH was obtained and stored at 4 °C.

### 4.2. UV Analysis

Aqueous solutions of TSP and TSPH at 1 mg/mL were scanned with a spectrophotometer (200–500 nm, UV-2900, Hengping, Shanghai, China).

### 4.3. TLC Analysis

The TLC analysis of TSPH solution (20 mg/mL) was performed on silica gel plate 5 × 10 cm (GF 254, Haiyang, China), activated in dry oven (100 °C, 8 min). The samples were unfolded with a mobile phase consisting of n-butanol (20 mL), acetic acid (20 mL) and water (10 mL). After unfolding, silica gel plates were developed in a color developer for 1 min, followed by heating at 100 °C until the bands were visible [60]. The color developer consisted of aniline (3 mL), diphenylamine (2 g), 85% phosphoric acid (10 mL), hydrochloric acid (1 mL), and of acetone (200 mL).

### 4.4. FT-IR Spectra Analysis

The infrared spectra of TSP and TSPH in the vibrational region 4000–500 cm^−1^ were measured with FT-IR Spectra (L1390021, PerkinElmer, Waltham, MA, USA).

### 4.5. HPLC–ESI/MS Analysis

The TSPH samples were first pre-column derivatized using PMP-methanol solution [61]. The TSPH solution (1 mg/mL, 50 μL) was mixed with NaOH solution (0.3 M, 50 μL) and PMP (0.5 M, 50 μL). The mixed solution was reacted at 80 °C for 0.5 h. After cooling, 50 μL of 0.3 M hydrochloric acid neutralization solution was added. Then 350 μL of chloroform was added to the solution and removed the organic phase after vortexing. The collected supernatant was filtered through a 0.22 μm microporous membrane (Merck KGaA, Darmstadt, Germany) and used for the HPLC–ESI/MS analysis [62,63]. More detailed experimental procedures are provided in the Appendix A.

### 4.6. Animal Experiments

A total of twenty SPF C57BL/6J mice (male; 6-weeks-old; body weight: 23–26 g) were purchased from Vital River Laboratory Animal Technology (Beijing, China). DSS (MW40000) was purchased from Aladdin Reagent (Shanghai, China). Mice were housed under a strict 12 h light cycle and the room was controlled at 22 ± 3 °C with 50 ± 5% humidity. All mice were fed standard food and sterile water.

At the end of the acclimatization feeding, the experiment was started (Figure 10). On the first day of the experiment, all mice were randomly assigned to 4 groups of 5 mice in each group (NOR, DSS, TSPH-L, TSPH-H). On days 1–7, sterile water for drinking in the DSS, TSPH-L, and TSPH-H groups was replaced with a 2.5% (*w*/*v*) sterile water solution of DSS. On days 8–15, aqueous TSPH solution was administered by gavage to mice in TSPH-L and TSPH-H groups at a dose of 2 g/kg/day and 4 g/kg/day, respectively. At the same time, mice in the NOR and DSS groups were administered sterile water in the same way. Daily weighing and detection of DAI during the experimental period. At the last day, a total of twenty mice were sacrificed while serum, cecum and colon contents, colon tissues, and other visceral organs were collected, then stored at −80 °C.

### 4.7. Evaluation of the Colon Histological

The fresh distal colon samples were fixed in Tissue-Tek^®^ OCT cryo embedding compound (Sakura Finetek, Torrance, CA, USA). After freezing overnight at −20 °C, colon tissues were cut into 10 μm sections and stained with H&E. The staining steps were performed as previous method with some modifications [64]. In short, the experimental protocol for staining was as follows: The obtained sections were fixed with 4% formaldehyde and rinsed with water. Then the sections were treated with hematoxylin solution, differentiation solution, and revertant blue solution for 1 min, respectively, and rinsed with distilled water for 20 s after each step. Slices were immersed in eosin solution for 30 s; after staining, the sections were rinsed with distilled water for 5 s, then dehydrated with 75%, 85%, 95% and 100% alcohol for 3 s, treated with absolute alcohol and xylene, respectively (1 min), and finally covered with gum and coverslips.

### 4.8. Measurement of Cytokines

Colon tissues (60 mg) were ground in 300 μL pre-chilled RIPA lysis buffer. The supernatant was collected after centrifugation (14,000× *g*; 5 min; 4 °C). The total colonic protein was measured by a BCA protein assay kit (Shanghai Beyotime Institute). After protein quantification, the content of cytokines in mouse serum and colonic tissues were measured by ELISA kits (Dakewe Biotechnology, Shenzhen, China).

### 4.9. Determination of TJ Proteins Related Gene Expression

Approximately 50 mg of colonic tissue was placed in a sterilized mortar (the mortar was pre-cooled with liquid nitrogen in advance) and grounded to powder by adding liquid nitrogen. The powder was collected in a 2 mL sterile centrifuge tube, 1 mL pre-cooled Trizol reagent was added, and RNA was extracted after thorough vortexing and homogenization. After reverse transcription of RNA to cDNA, real-time quantification was performed using ABScript III RT Master Mix (ABclonal, Wuhan, China). Relative values of the data were quantified according to the internal reference and calculated using the 2^−ΔΔCT^ method. The gene sequences used in this experiment and the PCR steps are provided in the Appendix A).

### 4.10. Assessment of TJ Proteins Expression

100 mg of frozen colon tissue was chopped up by surgical scissors and transferred to a precooled grinding tube. Two small sterilized steel balls were added, followed by 1 mM PMSF protease inhibitor and precooled RIPA lysis buffer solution, fully ground, and left on ice for 30 min at 4 °C. After centrifugation (3000× *g*, 15 min), all the supernatant was transferred to a 2 mL sterile centrifuge tube. The protein concentration was assayed using the BCA Protein Content Detection Kit, and then all sample proteins were diluted to a uniform concentration for subsequent experiments.

Western blot analysis was performed based on a previous study with some modifications [65]: A standard procedure was used to separate 15 ug colon tissue proteins from 12% SDS-PAGE gels. Proteins were transferred to the PVDF membrane using the wet transfer method, followed by incubation of the membrane in 30 mL of 5% skimmed milk for 1.5 h at room temperature. Primary antibodies Occludin (Rabbit, A2601), ZO-1 (Rabbit, A0659), β-actin (Rabbit, AC026) were used to incubate PVDF membranes overnight. After incubation with an HRP-conjugated secondary antibody (Goat, Ab6721) for 1 h at 4 °C, the PVDF membrane was subjected to signal detection using enhanced chemiluminescence (ECL) reagents (GE Amersham Imager 600, Cytiva, WA, USA). The strip gray values were quantified using Image J software V1.8.0.112 (Bethesda) and the relative protein expression using β-actin as an internal control. All of the antibodies were purchased from Wuhan ABclonal Biotechnology Co., Ltd. (Wuhan, China).

### 4.11. Determination of SCFAs

The concentration of SCFAs in colon contents was determined by reference to a previous study with slight modifications using a gas chromatograph-mass spectrometer (Agilent 7010B). The HP-FFAP-112 column (25 m × 0.32 mm × 0.5 μm) was used [66]. More detailed testing methods are provided in the Appendix A.

### 4.12. Gut Microbiota Analysis

Sample collection: During mouse dissection, cecal contents of mice were collected in cryopreserved tubes and stored to −80 °C refrigerator.

Total bacterial DNA was extracted from each sample using the CTAB method (cetyltrimethylammonium bromide method) according to the manufacturer’s instructions. The gene sequences used for subsequent PCR and more detailed experimental procedures are provided in the Appendix A.

### 4.13. Statistical Analysis

All graphical data were expressed as mean ± standard deviation (SD). One-way analysis of variance (ANOVA) and Tukey’s test were applied to analyze the data. *p* < 0.05 represents a significant difference between groups. The above work was performed using SPSS (Version 11.5) (*n* = 5 per group). All image processing was performed using GraphPad Prism 7 software.

## 5. Conclusions

TSPH is a newly identified hydrolyzed product of TSP containing mainly monosaccharides and oligosaccharides (DP = 2–4). TSPH can inhibit the overproduction of pro-inflammatory cytokines, enhance the expression of TJ proteins mRNA and protein expression, and further protect the intestinal barrier. TSPH uptake causes further colonization of SCFAs-producing bacteria such as *Blautia* and *Akkermansia* and decreases the abundance of *Coprobacillus*. All of these effects may have a positive protective effect on colitis. These data have developed new ideas for the utilization of TSPH and new insights into UC treatment and prevention.

## Figures and Tables

**Figure 1 pharmaceuticals-16-01133-f001:**
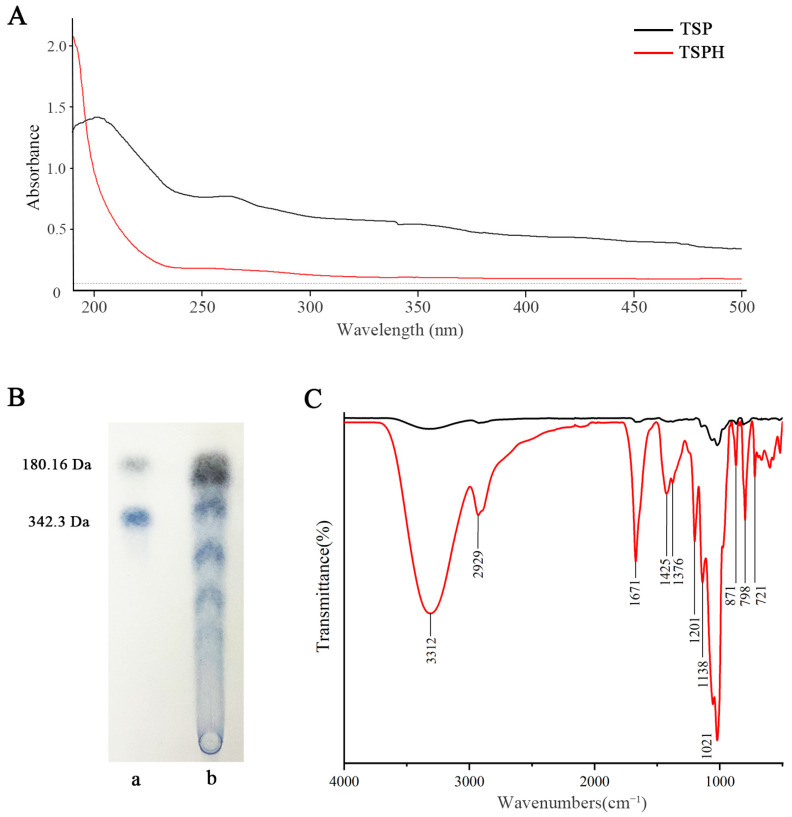
Structural composition analysis of TSPH. (**A**) ultraviolet spectroscopy (UV) analysis of TSP and TSPH. (**B**) Thin–layer chromatography (TLC) analysis of TSPH. (**C**) Fourier transform infrared spectroscopy (FT–IR) analysis of TSP and TSPH.

**Figure 2 pharmaceuticals-16-01133-f002:**
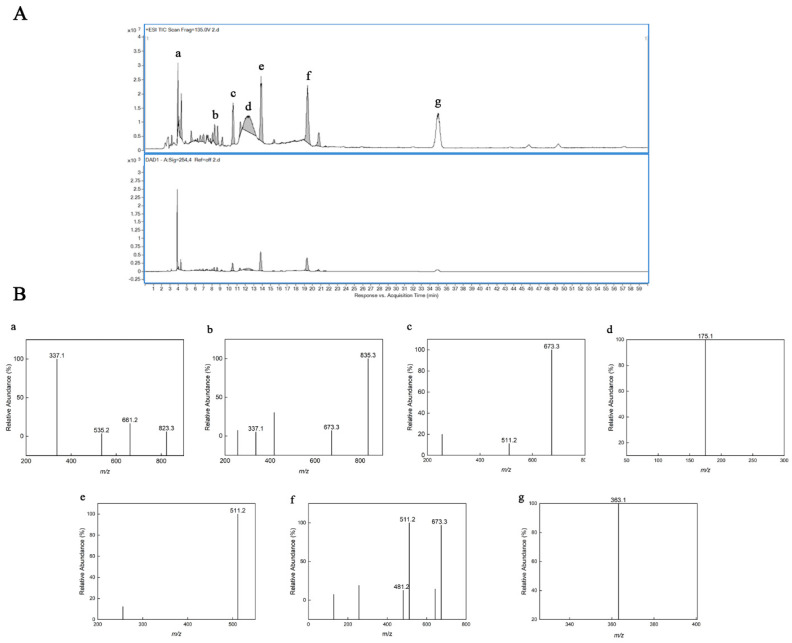
HPLC–ESI/MS of TSPH. (**A**) HPLC–UV chromatogram of TSPH. (**B**) ESI–MS analysis in positive mode of partially collected components (a–g) (**A**).

**Figure 3 pharmaceuticals-16-01133-f003:**
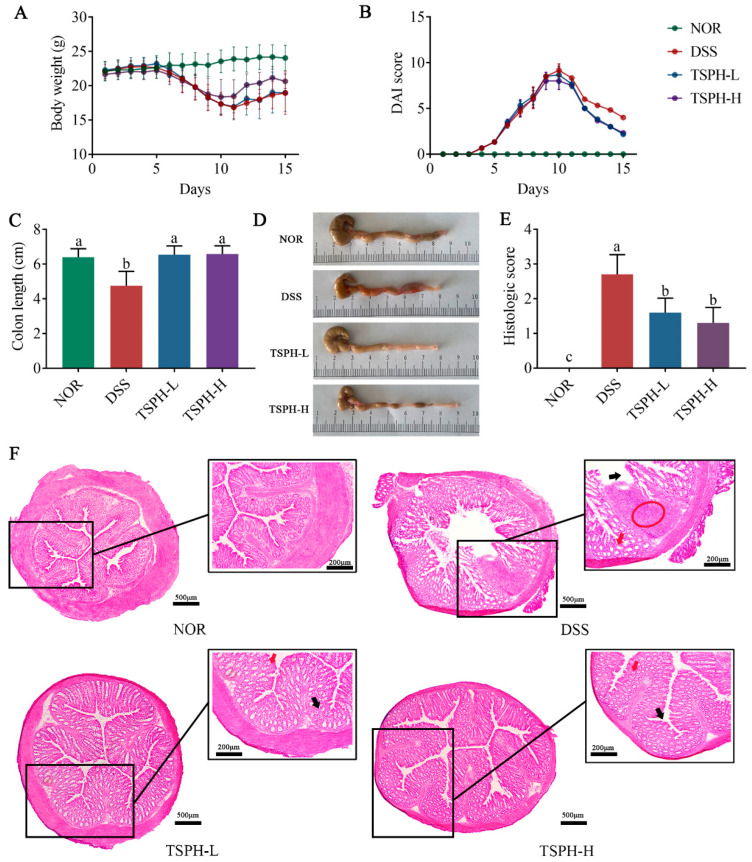
TSPH alleviated DSS−induced experimental colitis. (**A**) Daily body weight. (**B**) Disease activity index (DAI) scores. (**C**) The colon length of the mice was determined. (**D**) Photograph of a representative colon. (**E**) Histological disease score. (**F**) Distal colon sections stained with H&E. Red circles indicate inflammatory infiltrates, red arrows, and black arrows indicate chorionic surface and crypts, respectively. Data bars in the graph that do not share a common letter indicate that the difference between them is significant (*p* < 0.05).

**Figure 4 pharmaceuticals-16-01133-f004:**
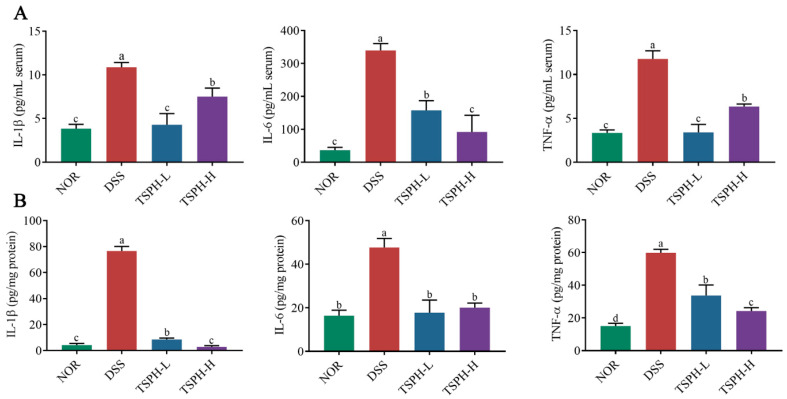
TSPH suppresses DSS-induced inflammatory activation. (**A**,**B**) The concentration of cytokines in serum and the colon. Data bars in the graph that do not share a common letter indicate that the difference between them is significant (*p* < 0.05).

**Figure 5 pharmaceuticals-16-01133-f005:**
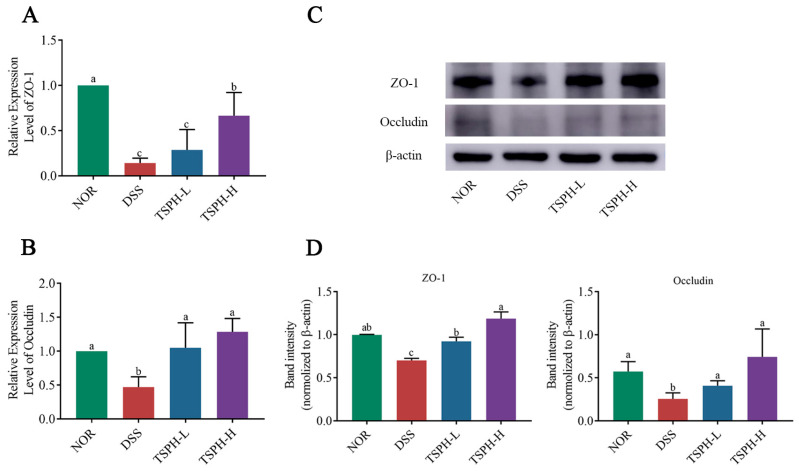
The effects of TSPH on TJ in colonic tissues were examined at the gene level and protein level, respectively. (**A**,**B**) The relative mRNA expression of ZO-1 and Occludin. (*n* = five per group). (**C**,**D**) The proteins expression of ZO-1 and Occludin. (*n* = three per group). Data bars in the graph that do not share a common letter indicate that the difference between them is significant (*p* < 0.05).

**Figure 6 pharmaceuticals-16-01133-f006:**
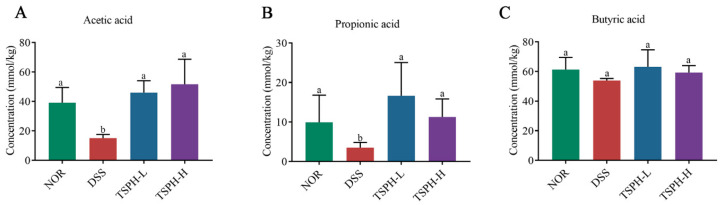
Effects of TSPH on the regulation of the content of SCFAs. (**A**–**C**): The change in acetic, propionic, and butyric acid. Data bars in the graph that do not share a common letter indicate that the difference between them is significant (*p* < 0.05). (*n* = five per group).

**Figure 7 pharmaceuticals-16-01133-f007:**
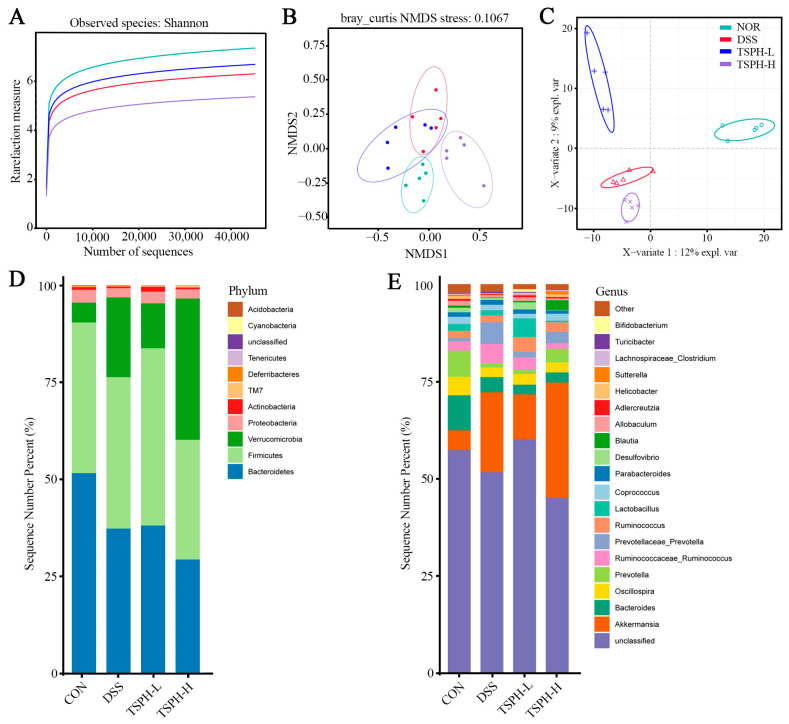
Effects of TSPH intestinal microbiota of UC mice. (**A**) Rarefaction Curve. (**B**,**C**) β−diversity was assessed by NMDS and PLS−DA. (**D**,**E**) The structural composition of microbiota on phylum and genus levels. (*n* = five per group).

**Figure 8 pharmaceuticals-16-01133-f008:**
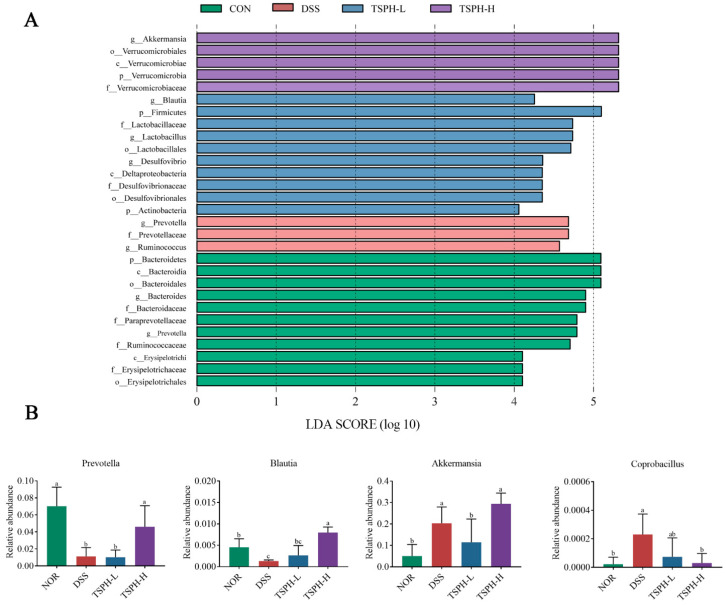
Effects of TSPH on microbiota in mice. (**A**) Histogram of the distribution of LDA values of microbial community differences. (**B**) The relative abundance of representative species, including *Prevotella*; *Blautia*; *Akkermansia*; *Coprobacillus*. Data bars in the graph that do not share a common letter indicate that the difference between them is significant (*p* < 0.05). (*n* = five per group).

**Figure 9 pharmaceuticals-16-01133-f009:**
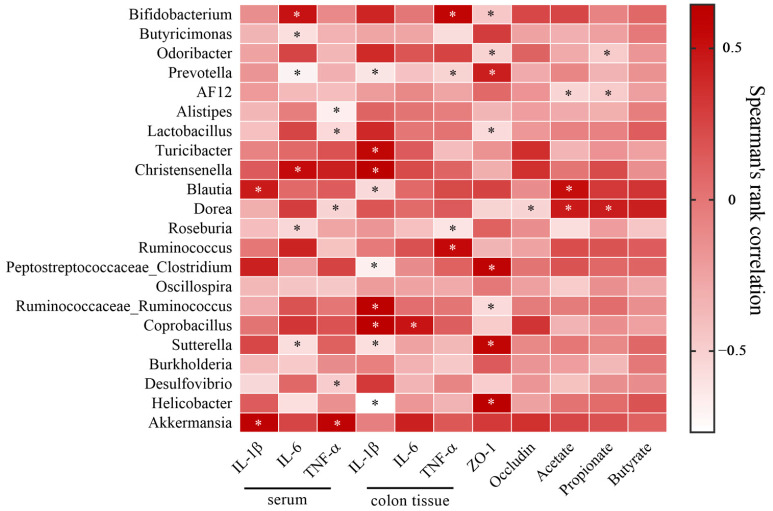
Correlation analysis between the intestinal bacteria and biochemical indexes. * *p* < 0.05. (*n* = five per group).

**Figure 10 pharmaceuticals-16-01133-f010:**
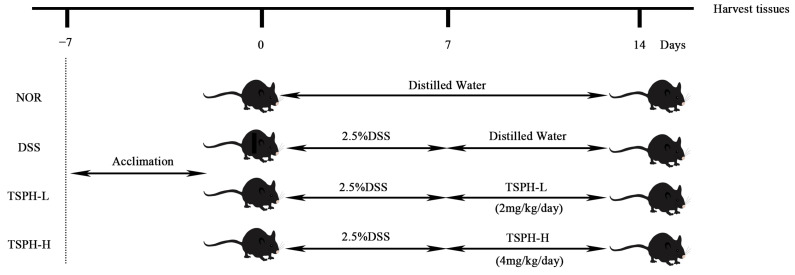
Diagram of animal experimental procedures.

## Data Availability

The cecum microbiota gene sequence dataset presented in this study has been made available at the NCBI online repository: NCBI; SubmissionID: SUB12343396; BioProject ID: PRJNA907084.

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
