# Peer review of "Tamarind Seed Polysaccharide Hydrolysate Ameliorates Dextran Sulfate Sodium-Induced Ulcerative Colitis via Regulating the Gut Microbiota"

_pharmaceuticals, 2023, doi:10.3390/ph16081133_

Round 1
Reviewer 1 Report
The authors reported how, Tamarind seed polysaccharide hydrolyzate can improve the inflammatory state that characterizes DSS-induced ulcerative colitis through the regulation of the intestinal microbiota.
The work is interesting, unfortunately there are some important critical issues that need to be resolved for the paper to be accepted.
1) Review the beginning of the introduction. An introductory sentence seems to be missing.
2) The introduction needs a revision of the English grammar.
3) The authors analyzed the p65 subunit of the transcription factor Nfkb. They should have analyzed the forphorylated form of p65 (pp65). The authors should have analyzed at least this to hypothesize the anti-inflammatory effect. This is a major criticality.
4) In my opinion, it would be preferable to also analyze the protein expression of ZO-1 and Occludin. mRNA expression analysis is not sufficient.
5) Indicate the antibody catalog number in the materials and methods section.
English needs review
Author Response
Dear Reviewer,
Thank you for your comments concerning our manuscript entitled “Tamarind seed polysaccharide hydrolysate ameliorates dextran sulfate sodium-induced ulcerative colitis via regulating the gut microbiota” (Manuscript ID: pharmaceuticals-2467953). The comments are all valuable and very helpful for revising and improving our paper, as well as the important guiding significance to our research. We have read the comments carefully and have made corrections which we hope meet with approval. Revised portions are marked in red in our revised manuscript. The main corrections in the paper and the responses to reviewer’s comments are as follows:
Reviewer Comments:
Reviewer: 1
Comments and Suggestions for Authors:
- The authors reported how, Tamarind seed polysaccharide hydrolyzate can improve the inflammatory state that characterizes DSS-induced ulcerative colitis through the regulation of the intestinal microbiota.
- The work is interesting, unfortunately there are some important critical issues that need to be resolved for the paper to be accepted.
Thank you for your summary. We really appreciate your efforts in reviewing our manuscript. We have revised the manuscript accordingly. Our point-by-point responses are detailed below.
- Review the beginning of the introduction. An introductory sentence seems to be missing.
Thank you for your comments. We have added introductory sentences to the "Introduction" section of the revised manuscript.
“Ulcerative colitis (UC) is a disease in which persistent tissue damage occurs as a result of inflammation affecting the colon or its entire mucosal surface to varying degrees[1]. UC was first described in 1859[2]” (Line 39-41)
- The introduction needs a revision of the English grammar.
Thank you for your careful review. We are very sorry for our poor writing in this manuscript and the inconvenience they caused in your reading. We tried our best to improve the manuscript and made some changes to the manuscript. And we hope the revised manuscript could be acceptable to you.
- The authors analyzed the p65 subunit of the transcription factor Nfkb. They should have analyzed the forphorylated form of p65 (pp65). The authors should have analyzed at least this to hypothesize the anti-inflammatory effect. This is a major criticality.
We greatly appreciate the reviewer’s comments. We strongly agree that analyzing the expression of phosphorylated p65 protein can better reveal the regulatory role of TSPH for NF-κB. However, in the present study, our primary focus was on the alleviating effect of TSPH on ulcerative colitis. In contrast, our subsequent study will focus on the mechanisms involved in alleviating colitis by TSPH, including the effects on the inflammatory pathway, and we will focus on the expression of pp65 protein. Therefore, we deleted this part of the study and added this deficiency in the "Discussion" section. We thank the reviewers for this valuable comment, which made this study more rigorous and provided us with new ideas for further research.
“There is growing evidence that DSS induces colitis due to the activation of the NF-κB inflammatory pathway by DSS, which would accelerate the secretion and accumulation of pro-inflammatory cytokines, leading to the development of colitis[3, 4]. These investigations provide an important future research direction to investigate whether TSPH exerts its anti-inflammatory effects by activating the NF-κB pathway.” (Line 253-258)
- In my opinion, it would be preferable to also analyze the protein expression of ZO-1 and Occludin. mRNA expression analysis is not sufficient.
We are very grateful to the reviewers for their valuable comments, which made this study more rigorous. We have added this part of experimental results and analysis in the manuscript.
Figure 5: The effects of TSPH on TJ in colonic tissues were examined at the gene level and protein level, respectively. (A-B): The relative mRNA expression of ZO-1 and Occludin. (n = 5 per group). (C-D): The proteins expression of ZO-1 and Occludin. (n = 3 per group). Data bars in the graph that do not share a common letter indicate that the difference between them is significant (P<0.05).
“Compared with the NOR group, DSS significantly decreased the transcript-level expression of ZO-1 and Occludin by 52.99% and 85.83% (P<0.01), respectively (Figure 5A-B). In colitis mice, the relative mRNA expression of ZO-1 was notably enhanced in the groups treated with 2 and 4 g/kg of TSPH (P<0.01). Data from Western Blot also confirmed this result (Figure 5C-D)” (Line 188-192).
- Indicate the antibody catalog number in the materials and methods section.
We deeply appreciate the reviewer’s suggestion. We have added the antibody catalog number in the materials and methods section in the revised manuscript.
“Occludin (Rabbit, A2601), ZO-1(Rabbit, A0659), β-actin (Rabbit, AC026) and HRP-conjugated secondary antibody (Goat, Ab6721)” (Line 389-391)
Thank you for your careful review. We really appreciate the effort you put into reviewing our manuscript. Your careful review has made our research clearer and more comprehensive.
[1] Wilks S, Morbid appearances in the intestine of Miss Bankes. [J].
[2] Khan S, Waliullah S, Godfrey V, et al., Dietary simple sugars alter microbial ecology in the gut and promote colitis in mice. [J]. Science Translational Medicine, 2020. 12(567).
[3] Neurath M F, Becker C, and Barbulescu K, Role of NF-kappaB in immune and inflammatory responses in the gut. [J]. Gut, 1998. 43(6): p. 856-60.
[4] Neurath M F, Fuss I, Schurmann G, et al., Cytokine gene transcription by NF-kappa B family members in patients with inflammatory bowel disease. [J]. Annals of the New York Academy of Sciences, 1998. 859: p. 149-59.

Reviewer 2 Report
This study is well-designed and performed. The results bring impact to the society.
Author Response
Dear Reviewer,
Thank you for your comments concerning our manuscript entitled “Tamarind seed polysaccharide hydrolysate ameliorates dextran sulfate sodium-induced ulcerative colitis via regulating the gut microbiota” (Manuscript ID: pharmaceuticals-2467953). The comments are all valuable and very helpful for revising and improving our paper, as well as the important guiding significance to our research. We have read the comments carefully and have made corrections which we hope meet with approval. Revised portions are marked in red in our revised manuscript. The main corrections in the paper and the responses to reviewer’s comments are as follows:
Reviewer: 2
Comments and Suggestions for Authors
This study is well-designed and performed. The results bring impact to the society.
We appreciate the reviewer’s positive evaluation of our work.
Thank you for your careful review. We really appreciate the effort you put into reviewing our manuscript. Your careful review has made our research clearer and more comprehensive.

Reviewer 3 Report
The submitted article entitled: Tamarind seed polysaccharide hydrolysate ameliorates dextran sulfate sodium-induced ulcerative colitis via regulating the gut microbiota, presents interesting and very useful research on the functional and health benefits of Tamarind. The introduction is well and clear written. In the results, the effects of TSPH on microbiota is clearly demonstrated. The statistical analysis is elaborative and clear, including the heat map which clearly shows the correlation coefficient between bacterial taxa and colitis-related index.
The following are few comments which should be addressed before further steps:
1-Introduction:
line 36: remove the full stop at the end of the first sentence or use a suitable connector.
2-Materials:
please explain more why these concentrations were specifically used:
2.5% DSS
2mg TSPH-L
4 mg TSPH-H
Author Response
Dear Reviewer,
Thank you for your comments concerning our manuscript entitled “Tamarind seed polysaccharide hydrolysate ameliorates dextran sulfate sodium-induced ulcerative colitis via regulating the gut microbiota” (Manuscript ID: pharmaceuticals-2467953). The comments are all valuable and very helpful for revising and improving our paper, as well as the important guiding significance to our research. We have read the comments carefully and have made corrections which we hope meet with approval. Revised portions are marked in red in our revised manuscript. The main corrections in the paper and the responses to reviewer’s comments are as follows:
Reviewer 3:
Comments and Suggestions for Authors
- The submitted article entitled: Tamarind seed polysaccharide hydrolysate ameliorates dextran sulfate sodium-induced ulcerative colitis via regulating the gut microbiota, presents interesting and very useful research on the functional and health benefits of Tamarind. The introduction is well and clear written. In the results, the effects of TSPH on microbiota is clearly demonstrated. The statistical analysis is elaborative and clear, including the heat map which clearly shows the correlation coefficient between bacterial taxa and colitis-related index.
- The following are few comments which should be addressed before further steps:
Thank you for your summary. We really appreciate your efforts in reviewing our manuscript. We have revised the manuscript accordingly. Our point-by-point responses are detailed below.
1-Introduction:
line 36: remove the full stop at the end of the first sentence or use a suitable connector.
Thank you for your comments and professional advice. We agree with the comment and rewrote the sentence in the revised manuscript as the following:
“Ulcerative colitis (UC) is a disease in which persistent tissue damage occurs as a result of inflammation affecting the colon or its entire mucosal surface to varying degrees[1]. UC was first described in 1859[2]” (Line 39-41)
2-Materials:
please explain more why these concentrations were specifically used: 2.5% DSS, 2mg TSPH-L, 4 mg TSPH-H.
Thank you for your comments. Our explanation is as follows:
- The concentration of DSS:5%.
In recent decades, a variety of animal models have been established to study the possible mechanisms of UC pathogenesis and therapeutic approaches, and these modeling methods mainly include chemical induction, bacterial induction, and genetic modification, with chemical induction methods being the most commonly used. The dextran sodium sulfate (DSS) modeling method is simple, fast, and reproducible, which can simulate the whole process of UC occurrence and development, and its lesion symptoms are very similar to the clinical symptoms of human UC[3, 4]. Therefore, DSS solution was chosen to induce mice to establish UC mouse model in this study.
Based on the concentrations used in previous studies, we set up three concentrations of DSS solutions, 1%, 2.5%, and 5%, for acute colitis mouse modeling in the pre-experiment[5-8]. During the pre-experiment, it was observed that mice in the 1% DSS group showed insignificant signs of enteritis, presumably due to the low concentration of DSS. 2.5% DSS and 5% DSS groups showed significant shortening of the length of the colon, and HE staining showed more significant inflammatory cell infiltration. However, mice in the 5% group showed more severe deaths during the experiment, presumably due to the high concentration of DSS. Therefore, in the subsequent formal experiments, we chose 2.5% DSS solution for UC modeling.
- The concentration of TSPH: TSPH-L: 2mg/kg/day; TSPH-H: 4mg/kg/day.
In order to more appropriately transfer drug doses from animal to human studies, Shaw et al. suggested using the body surface area (BSA) normalization method. The conversion factor varies between animals and humans according to the BSA method. The dose administered to mice is approximately 12.3 times that of humans[9]. In our pre-experiment, we chose ChangYanNing(CYN) as a positive control. CYN is China's most common drug to relieve chronic enteritis[10]. In addition, various natural components of CYN, such as Euphorbia humifusa Willd, have been reported to have some anti-inflammatory effects[11]. Thus, CYN was chosen as the positive control in the pre-experiment.
The human dose of CYN is 166.7 mg/kg/day and the dose administered to mice is approximately 12.3 times the human dose, so we chose 2 g/kg/day as the concentration of TSPH-L and 4 mg/kg/day as the concentration of TSPH-H.
Thank you for your careful review. We really appreciate the effort you put into reviewing our manuscript. Your careful review has made our research clearer and more comprehensive.
[1] Wilks S, Morbid appearances in the intestine of Miss Bankes. [J].
[2] Khan S, Waliullah S, Godfrey V, et al., Dietary simple sugars alter microbial ecology in the gut and promote colitis in mice. [J]. Science Translational Medicine, 2020. 12(567).
[3] Wirtz S, Neufert C, Weigmann B, and Neurath M F, Chemically induced mouse models of intestinal inflammation. [J]. Nat Protoc, 2007. 2(3): p. 541-6.
[4] Jialing L, Yangyang G, Jing Z, et al., Changes in serum inflammatory cytokine levels and intestinal flora in a self-healing dextran sodium sulfate-induced ulcerative colitis murine model. [J]. Life Sci, 2020. 263: p. 118587.
[5] Zhao B, Wu J, Li J, et al., Lycopene Alleviates DSS-Induced Colitis and Behavioral Disorders via Mediating Microbes-Gut-Brain Axis Balance. [J]. J Agric Food Chem, 2020. 68(13): p. 3963-3975.
[6] Wei Y Y, Fan Y M, Ga Y, Zhang Y N, Han J C, and Hao Z H, Shaoyao decoction attenuates DSS-induced ulcerative colitis, macrophage and NLRP3 inflammasome activation through the MKP1/NF-κB pathway. [J]. Phytomedicine, 2021. 92: p. 153743.
[7] Jiang N, Wei Y, Cen Y, et al., Andrographolide derivative AL-1 reduces intestinal permeability in dextran sulfate sodium (DSS)-induced mice colitis model. [J]. Life Sci, 2020. 241: p. 117164.
[8] Zhang X J, Yuan Z W, Qu C, et al., Palmatine ameliorated murine colitis by suppressing tryptophan metabolism and regulating gut microbiota. [J]. Pharmacol Res, 2018. 137: p. 34-46.
[9] Reagan-Shaw S, Nihal M, and Ahmad N, Dose translation from animal to human studies revisited. [J]. Faseb Journal, 2008. 22(3): p. 659-661.
[10] Yang X, Yang S P, Zhang X, Yu X D, He Q Y, and Wang B C, Study on the Multi-marker Components Quantitative HPLC Fingerprint of the Compound Chinese Medicine Wuwei Changyanning Granule. [J]. Iranian Journal of Pharmaceutical Research, 2014. 13(4): p. 1191-1201.
[11] Bui Thi Thuy L, Bui Huu T, Nguyen Phuong T, et al., Anti-inflammatory components of Euphorbia humifusa Willd. [J]. Bioorganic & Medicinal Chemistry Letters, 2014. 24(8): p. 1895-1900.
